Inclusion of up to 20% Black Soldier Fly larvae meal in broiler chicken diet has a minor effect on caecal microbiota

de Souza Vilela Jessica 1 jv.zootecnia@gmail.com
Kheravii Sarbast K. 1
http://orcid.org/0000-0002-3043-071X Sharma Bajagai Yadav 2
http://orcid.org/0000-0002-5999-0374 Kolakshyapati Manisha 1
Zimazile Sibanda Terence 1
Wu Shu-Biao 1
http://orcid.org/0000-0002-2850-2307 Andrew Nigel R. 1 3
http://orcid.org/0000-0001-5423-9306 Ruhnke Isabelle 1
1 School of Environmental and Rural Science, University of New England , Armidale, NSW , Australia
2 Institute for Future Farming Systems, Central Queensland University , Rockhampton, Queensland , Australia
3 Faculty of Science and Engineering, Southern Cross University , Lismore, NSW , Australia
Oppert Brenda
Electronic publication date: 2023 Sep 19
Publication date: 2023
Volume: 11
Electronic Location ID: e15857
Received 2023 Feb 28; Accepted 2023 Jul 16
Copyright: © 2023 de Souza Vilela et al.
Copyright year: 2023
Copyright holder: de Souza Vilela et al.
License: This is an open access article distributed under the terms of the Creative Commons Attribution License, which permits unrestricted use, distribution, reproduction and adaptation in any medium and for any purpose provided that it is properly attributed. For attribution, the original author(s), title, publication source (PeerJ) and either DOI or URL of the article must be cited.
License URL: https://creativecommons.org/licenses/by/4.0/

Keywords: Hermetia illucens, Insects, Broiler chickens, Microbiota, Poultry

Funding: Poultry Hub Australia 18-409 This work was supported by Poultry Hub Australia (project number 18-409). The funders had no role in study design, data collection and analysis, decision to publish, or preparation of the manuscript.

==============================
Background

The Black Soldier Fly larvae (BSFL) are a source of nutrients and bioactive compounds in broiler diets. Some components of the BSFL may serve as a prebiotic or may impact the intestinal microbiota of the broilers by other modes of action, which in turn can affect the health and performance of broilers. Here, we investigate the impact of up to 20% BSFL in broiler diets on the diversity and composition of the broiler’s microbiota.

Methods

Four hundred broilers were fed five iso-nutritious experimental diets with increasing levels of BSFL meal reaching 0%, 5%, 10%, 15%, 20% in the finisher diets. Eight caecal content samples coming from each of the eight replicates per treatment were collected at two time points (day 21 and day 42) for DNA extraction and sequencing of the V3–V4 regions using Illumina MiSeq 2 × 300 bp pair-end sequencing with 341f and 805r primers. Analysis of variance and Spearman’s correlation were performed, while QIIME2, DADA2, and Calypso were used for data analysis.

Results

When broilers were 21 days of age, the abundance of two groups of sequence variants representing Enterococcus and unclassified Christensenellaceae was significantly lower (p-value = 0.048 and p-value = 0.025, respectively) in the 20% BSFL group compared to the 0% BSFL group. There was no relevant alteration in the microbiota diversity at that stage. On day 42, the Spearman correlation analysis demonstrated that the sequence variants representing the genus Coprococcus showed a negative relationship with the BSFL inclusion levels (p-value = 0.043). The sequence variants representing the genus Roseburia and Dehalobacterium demonstrated a positive relationship with the BSFL dietary inclusion (p-value = 0.0069 and p-value = 0.0034, respectively). There was a reduction in the dissimilarity index (ANOSIM) caused by the 20% BSFL dietary inclusion.

Conclusion

The addition of up to 20% BSFL in broiler diets did not affect the overall caeca microbiota diversity or composition at day 21. On day 42, there was a reduction in the beta diversity caused by the 20% BSFL dietary inclusion. The abundance of the bacterial group Roseburia was increased by the BSFL dietary inclusion, and it may be beneficial to broiler immunity and performance.

Introduction

In the context of climate change and its effects, insect-based diets can be a sustainable option to improve resource and ingredient uses in animal diets (Gasco et al., 2019). If insect meals could be produced on a large scale considering sustainable growing methods, the use of insect meals may address the predicted future shortage of conventional protein ingredients currently used in livestock feed (de Souza-Vilela, Andrew & Ruhnke, 2019).

Some insects can be fed with organic waste using fewer natural resources including less water, energy and landmass, facilitating insect production even in conditions known for its limited productivity (Madau et al., 2020). The Black Soldier Fly larvae (BSFL; Hermetia illucens) is one of the insect species known for its suitability for mass production and a nutrient composition allowing it to be used especially in monogastric animal feed (Alagappan et al., 2022; Chia et al., 2018; de Souza-Vilela et al., 2021). The BSFL also has the potential to reduce the amount of human waste production as it can be grown on various organic materials and result in a feed product with up to 79% crude protein content and a crude fat content of up to 58% on dry matter basis (de Souza-Vilela et al., 2021; Lu et al., 2021; Hopkins et al., 2021; Lu et al., 2022). The BSFL is also known for containing a variety of microbiota modulating substances such as chitin, lauric acid, and up to 57 different antimicrobial peptides (de Souza-Vilela, Andrew & Ruhnke, 2019; Jantzen et al., 2020). Diet-dependent genes encoding putative antimicrobial peptides in the BSFL have defensive power against a broad range of bacteria (Moretta et al., 2020; Vogel et al., 2018). Chitin content varies with the rearing substrate and BSFL life stage. The chitin content has been reported to be 5.6% to 6.7% on a dry matter basis, depending on the rearing substrate (Spranghers et al., 2017). The BSFL lauric acid content varies between 13.4% and 51.8% on a dry matter basis, depending on the rearing substrate (e.g., ensiled mussels and bread, respectively) (Ewald et al., 2020). However, while these nutrient levels can vary, the growing conditions of the BSFL can be tightly controlled and monitored. This allows for a real advantage compared to conventional feed materials such as grains where nutrient levels commonly vary due to weather, soil, harvesting conditions and others.

The gastrointestinal tract (GIT) microbiota of poultry is associated with the bird’s nutrition, health, brain functioning, behavior, immunity, performance, and welfare status (Kraimi et al., 2019; Rubio, 2019; Valdes et al., 2018). The GIT microbiome plays a crucial role in the efficiency of production animals, such as broiler chickens as it can increase the energy metabolism of the host, directly or indirectly modulate the immune system, determine the impact of various pathogens, and directly influence economic success for the farmers (Diaz Carrasco, Casanova & Fernández Miyakawa, 2019; Fagundes et al., 2017; Józefiak et al., 2020; Foysal et al., 2019; Gasco, Finke & Van Huis, 2018; Lei et al., 2019; Roos & Van Huis, 2017; Spranghers et al., 2018; Lopez-Santamarina et al., 2020; Sivieri et al., 2017; Wu et al., 2021).

In vivo and in vitro studies reported that both the whole BSFL and it’s isolated antimicrobial compounds (e.g., chitin, antimicrobial peptide, lauric acid) affected the GIT microbiota of poultry, humans, and mice by various modes of action (Borrelli et al., 2017; Lopez-Santamarina et al., 2020; Moll et al., 2022; Selenius et al., 2018). For example, Borrelli et al. (2017) reported that the dietary inclusion of 17% defatted BSFL (containing 4.6% fat and 2.2% chitin) in 45-weeks-old Lohmann Brown Classic laying hens increased the abundance of beneficial bacteria, including Flavonifractor plautii, Christensenella minuta, and Alkaliphilus transvaalensis in layer caeca. Even a low concentration of 0.01% of chitin in an in vitro experiment demonstrated a powerful antibacterial effect and completely inhibited the growth of the harmful strain of the bacteria Escherichia coli TG (Selenius et al., 2018). Similarly, Yang et al. (2019) demonstrated through sequencing that 400 mg lauric acid/kg diet feeding broilers (Cobb strain) for 11 days, results in a significant decrease in the abundance of the pathogen Escherichia Shigella in the broiler’s jejunum.

Improvements in the broiler’s GIT microbiota can also positively affect animal health and welfare by reducing symptoms of otherwise costly diseases. Necrotic enteritis is an example of an infectious disease mainly caused by pathogenic Clostridium perfringens and is well known to be responsible for financial losses in the world’s poultry industry (Shojadoost et al., 2022). Modification of broilers’ GIT microbiota has been demonstrated to mitigate clinical symptoms of necrotic enteritis. For example, the oral dosage of 107 and 108 colony-forming units (CFU) of Lactobacilli per broiler increased the quantity of gram-positive bacterial taxa such as Actinobacteria, Lactobacillaceae, and Firmicutes in the intestine of broilers accompanied by reduced clinical signs of necrotic enteritis (Shojadoost et al., 2022). Similarly, results by Zaghari, Sarani & Hajati (2020) showed that the oral administration of the gram-positive Bacillus licheniformis (109 CFU/g, 0.5 g/kg diet) improved broiler performance, production efficiency, and return on investment compared to birds of the control group.

As a food source, BSFL modified the gastrointestinal microbiota of the host directly via its antimicrobial substances (Çetingül & Shah, 2022; Vogel et al., 2018). The challenge with most experiments investigating various levels of microbiota-modulating ingredients including BSFL is that the ingredients are commonly excessive causing dilution of other nutrients in the diet. This can lead to imbalanced diets and varying feed intake levels, adjusting for the different energy densities, of the experimental diets. Therefore, the aim of this experiment was to investigate the potential benefits of the microbiota-modulating feed ingredient, Black Soldier Fly larvae, when used at various inclusion levels allowing for a balanced and complete diet meeting the dietary recommendations of commercial broiler breeds.

The objective of this study was to evaluate the impact of up to 20% full-fat BSFL meal implemented in a balanced, iso-caloric and iso-nitrogenous diet on the composition and diversity of the GIT microbiome at genus and operational unit taxonomic in broilers. We expected that four dietary levels of BSFL would alter the cecal microbiota of broilers at 21 and 42 days of age.

Materials and Methods

Experimental design and intestinal sampling

The dried full-fat BSFL was obtained from a commercial producer (Karma3, Thomastown, VIC, Australia). The BSFL was reared on commercial poultry feed, harvested on day 12, and dried at 80 °C. The BSFL and all other dietary raw materials were analysed by standard methods (see de Souza-Vilela et al., 2021) for their nutrient composition and then mixed using a feed mixer (GD Safemix; GD Engineering, Merrylands, NSW, Australia) before being pelleted (PP300SW; Palmer Milling, Griffith, NSW, Australia). The nutrient composition of the diets varied between the broiler’s age and phases (starter, grower, finishers) as demonstrated in Table 1. The diets were formulated based on the nutrient requirements of the commercial breed (Ross 308) (Aviagen, 2016). Based on a randomized complete design, four hundred male Ross-308 broilers were assigned to five dietary treatment groups, each with eight replicates. Eight samples per treatment of each time point were collected based on the analysis performed. Broilers were housed in cages with wire mesh floors, preventing broilers from accessing their excreta. All broilers were housed in climate control rooms and received the following experimental diets ad libitum: the ‘starter’ diet (day 2 to day 10 of age) included 0%, 2.5%, 5%, 7.5%, and 10% BSFL; the ‘grower’ diet (day 11 to day 21 of age) and the ‘finisher’ diet (day 22 to day 42 of age) included 0%, 5%, 10%, 15%, and 20% of BSFL. Sick animals were culled and subtracted from the experiment. All diets were formulated to meet or exceed the nutrient specifications as outlined by the breeder (Aviagen, 2016). Environmental conditions such as temperature, feed and water availability were measured at least three times a day in all experimental units to ensure equal conditions and avoid confounders.

Table 1 Ingredient composition of diets in different stages; starter, grower, and finisher diets.

	Starter diets	Grower diets	Finisher diets	
Ingredients (%)	T1	T2	T3	T4	T5	T1	T2	T3	T4	T5	T1	T2	T3	T4	T5	
Wheat grain	53.4	53.9	54.2	53.5	52.8	59.0	59.4	58.7	55.4	54.5	64.1	63.4	62.1	60.7	59.5	
Soybean meal	33.5	31.4	29.6	28.6	27.6	28.0	26.1	24.4	22.7	20.7	23.5	21.4	20.1	18.7	16.4	
BSFL	0.00	2.50	5.00	7.50	10.0	0.00	5.00	10.0	15.0	20.0	0.00	5.00	10.0	15.0	20.0	
Canola oil	3.16	2.37	1.63	1.02	0.41	4.29	2.88	1.83	1.52	0.72	0.00	0.00	0.00	0.00	0.00	
Cottonseed oil	0.00	0.00	0.00	0.00	0.00	0.00	0.00	0.00	0.00	0.00	5.53	4.28	3.08	1.88	0.53	
Meat and bone meal	3.00	3.00	3.00	3.00	3.00	3.72	2.00	0.51	1.31	0.00	2.51	1.71	0.89	0.07	0.47	
Hulled oat	3.00	3.00	3.00	3.00	3.00	0.00	0.00	0.00	0.00	0.00	0.00	0.00	0.00	0.00	0.00	
Celite	2.00	2.00	2.00	2.00	2.00	2.00	2.00	2.00	2.00	2.00	2.00	2.00	2.00	2.00	2.00	
Limestone	0.98	0.84	0.68	0.52	0.00	0.89	0.84	0.76	0.39	0.28	0.88	0.72	0.56	0.39	0.00	
CaHPO4	0.05	0.00	0.00	0.00	0.00	0.00	0.00	0.00	0.00	0.00	0.00	0.00	0.00	0.00	0.00	
Salt	0.22	0.21	0.19	0.20	0.21	0.06	0.19	0.19	0.18	0.16	0.14	0.23	0.21	0.20	0.15	
Na bicarbonate	0.00	0.00	0.00	0.00	0.00	0.15	0.15	0.15	0.15	0.15	0.15	0.15	0.15	0.15	0.15	
TiO2	0.00	0.00	0.00	0.00	0.00	0.50	0.50	0.50	0.50	0.50	0.50	0.50	0.50	0.50	0.50	
Vitamin premix	0.05	0.05	0.05	0.05	0.05	0.05	0.05	0.05	0.05	0.05	0.05	0.05	0.05	0.05	0.05	
Mineral premix	0.08	0.08	0.08	0.08	0.08	0.08	0.08	0.08	0.08	0.08	0.08	0.08	0.08	0.08	0.08	
Phytase	0.01	0.01	0.01	0.01	0.01	0.01	0.01	0.01	0.01	0.01	0.01	0.01	0.01	0.01	0.01	
Choline	0.06	0.07	0.08	0.08	0.08	0.05	0.06	0.07	0.08	0.09	0.05	0.06	0.06	0.07	0.08	
L-Lysine HCl (78.4)	0.19	0.19	0.18	0.15	0.01	0.66	0.30	0.26	0.19	0.16	0.18	0.14	0.07	0.01	0.00	
D, L-Methionine	0.34	0.34	0.33	0.31	0.30	0.35	0.32	0.3	0.28	0.27	0.23	0.21	0.19	0.16	0.13	
L-Isoleucine	0.00	0.00	0.00	0.00	0.00	0.08	0.06	0.04	0.00	0.00	0.01	0.00	0.00	0.00	0.00	
L-Arginine	0.00	0.00	0.00	0.00	0.00	0.11	0.13	0.15	0.17	0.17	0.00	0.00	0.00	0.00	0.00	
Total	100	100	100	100	100	100	100	100	100	100	100	100	100	100	100	
Notes:

*The diets were iso-caloric and iso-nitrogenous (de Souza-Vilela et al., 2021).

Each data point represents the amount of each ingredient at each treatment within each stage (starter, grower, finisher).

Details regarding the BSFL and diet composition can be found in de Souza-Vilela et al. (2021). Briefly, the full-fat BSFL had 40% protein, 32.5% fat, and 13% lauric acid on a dry matter basis. As the BSFL inclusion levels increased in the diets, the lauric acid content also gradually increased from 0% lauric acid (control diet), 0.65% lauric acid (5% BSFL inclusion diet), 1.3% lauric acid (10% BSFL inclusion diet), 1.95% lauric acid (15% BSFL inclusion diet), and 2.6% lauric acid (20% BSFL inclusion diet).

However, the total dietary fat level was maintained comparable between the treatment groups. The formulation and ingredient composition of the treatment diets are detailed in Table 1. The lauric acid content which was determined through the fatty acid content. The fatty acid composition of the BSFL was determined according to the method described by Clayton et al. (2012). Based on the literature, we estimated the chitin content between 6% and 7% as the BSFL used in our experiment was reared on a conventional chicken feed (Spranghers et al., 2017). The chitin content also gradually increased as the level of BSFL increased in the experimental diets, being 0%, 0.3%, 0.6%, 0.9%, and 1.2% in control, 5%, 10%, 15%, and 20% BSFL inclusion diets. The antimicrobial peptide content present in the BSFL used in this study was not identified.

At the end of the grower and finisher phase, when broilers were 21 and 42 days old, one broiler per cage (one cage representing a statistical unit) was individually weighted (SB32001 DeltaRange Balance; Mettler Toledo, Painesville, OH, USA), electrical stunned (Stunner; JF, Weltevreden, South Africa), decapitated and degutted, and then the caeca content samples were collected into sterile 2 mL Eppendorf tubes (3810X; Eppendorf AG, Hamburg, Germany). All samples were snap-frozen with liquid nitrogen and then stored at −80 °C until required for the DNA extraction. The sample sizes were calculated based on the population size, a confidence interval of 95%, and a margin of error of 5%. The University of New England Animal Ethics Committee approved this research (AEC18-084).

DNA extraction

The caeca DNA was extracted using DNeasy PowerSoil Pro kit (Qiagen, Inc., Doncaster, VIC, Australia) and as described by Kheravii et al. (2017) with some modifications. In brief, approximately 300 mg of glass beads (0.1 mm) and 60 mg of frozen caeca contents were placed in 1.5 mL Eppendorf tubes. Then, 800 μl of solution CD1 was pipetted to samples prior to the disruption of the cells by Tissuelyser II (Qiagen, Inc., Hilden, Germany) for 5 min at a frequency of 30/s. The samples were incubated at 90 °C for 10 min and then centrifuged at 20,000 × g for 5 min. The supernatant was transferred onto a loading block, and the DNA extraction was performed using the QIAcube HT instrument (Qiagen, Hilden, Germany) following the manufacturer’s instructions. The caeca DNA was diluted in autoclaved water to reach the desired concentration of approximately 8 ng/µl.

DNA sequencing and data analysis

The V3–V4 regions of the 16S rRNA gene were sequenced using Illumina MiSeq 2 × 300 bp pair-end sequencing with 341f and 805r primers. The quality of the sequence reads was checked with fastQC v0.11.9 (Andrews, 2010). The upstream analysis of the sequence was done with QIIME2 (Bolyen et al., 2019) using DADA2 (Callahan et al., 2016) plugin for quality control and denoising. The downstream statistical analysis of the amplicon sequence variants (ASV) matrix was done with the online software, Calypso (Zakrzewski et al., 2016). All data were normalized using SquareRoot transformation, performed for day 21 and day 42, at genus and Operational Taxonomic Units (OTU) levels. The primary group as biological condition was used to analyse the data for all analyses (Chao1, Shannon Index, ANOSIM of Bray-Curtis, PCA, and abundance of amplicon sequence variants). Samples were rarefied to read depth of 15,730 at day 21, and 13,063 at day 42.

Results

Sequence quality control

A total of approximately 1.8 million quality-filtered sequences were obtained after filtering, denoising, chimeral removal, and singleton and doubleton removal. The minimum and maximum sequences per sample were 3,606 and 37,749, respectively. The data were rarefied to the minimum number of sequences for downstream analysis.

Overview of microbiota profile

As anticipated, notable differences were observed between the microbiota profiles of birds on day 21 and day 42 (Fig. 1). On day 21, the predominant bacteria belonged to the phylum Firmicutes, accounting for over 90% of the microbial population in all the treatment groups. Additionally, phyla Firmicutes, Proteobacteria, and Tenericutes collectively constituted almost 100% of the bacterial population. On day 42, while Firmicutes remained the most dominant phylum, its proportion was lower compared to day 21. Firmicutes, Bacteroides, and Proteobacteria constituted more than 90% of the microbiota population in all treatment groups on day 42. The most significant change at the phylum level between day 21 and day 42 was the substantial increase in the proportion of Bacteroidetes on day 42. At day 21, the top three genera were Faecalibacterium, Lactobacillus, and Ruminococcus, whereas, on day 42, Bacteroides, Faecalibacterium, and Barnesiella were the dominant genera across all treatment groups. Figure 1 displays the top 20 genera in all treatment groups at both sampling points, day 21 and day 42.

Figure 1 Stacked bar plots showing general microbiota profile at phylum and genus level.

(A) Phylum level taxa at day 21. (B) Phylum level taxa at day 42. (C) Top 20 genera at day 21. (D) Top 20 genera at day 42. T1 (20%), T2 (15%), T3 (10%), T4 (5%), and T5 (0%).

Alpha diversity in the caeca microbiome

The box plots (Fig. 2) show the assessment of the Chao1 microbial richness (the least count of OTUs in a sample) index at the amplicon sequence variants level, indicating that there were no significant differences within the samples from the caeca of broilers fed with grading levels of BSFL either at day 21 (p-value = 0.28; Fig. 2A) or at day 42 (p-value = 0.21; Fig. 2B). The violin plots (Fig. 3) represent the results of Shannon index, where no significant difference was observed in the bacterial diversity in the caeca of broilers fed increasing levels of BSFL either at day 21 (p-value = 0.18; Fig. 3A) or at day 42 (p-value = 0.71; Fig. 3B) compared to the Shannon diversity index of broilers not fed with BSFL (control group).

Figure 2 (A) Chao1 diversity of 21 days old broilers fed 0%, 5%, 10% 15% or 20% Black Soldier Fly larvae (BSFL). (B) Chao1 diversity of 42 days old broilers fed 0 (red), 5 (green), 10 (blue), 15 (grey) or 20% (yellow) BSFL.

Box plots representing the variation of the Chao1 diversity (alpha diversity) index.

Figure 3 (A) Shannon index at genus level of 21 days old broilers fed 0%, 5%, 10%, 15%, or 20% BSFL. (B) Shannon index at amplicon sequence variant (ASV) level of 42 days old broilers fed 0%, 5%, 10%, 15%, or 20% BSFL.

Shannon Index in broilers caeca microbiota of different groups was compared through ANOVA analysis.

Beta diversity in the caeca microbiome

Analysis of similarities (ANOSIM Bray-Curtis) showed no difference in the microbiome structure in the caeca of broilers fed increasing levels of BSFL either at ASV or at the genus level for 21 days old broilers (p-value = 0.561; Fig. 4A). However, in the caeca of 42 days old broilers, there was a significant difference in the indexes of dissimilarity at OTU level (p-value = 0.022; Fig. 4B). The caeca microbiota community profiles of 21- and 42-days old broilers were grouped by hierarchical clustering and ordinated by principal component analysis (PCA), and as shown in Fig. 5, there were no differences between treatments.

Figure 4 (A) ANOSIM Bray-Curtis (genus level) in the caeca of 21 days old broilers fed 0 (red), 5 (green), 10 (blue), 15 (grey), or 20% (yellow) Black Soldier Fly larvae (BSFL).

(B) ANOSIM Bray-Curtis at in the caeca of 42 days old broilers fed 0%, 5%, 10%, 15%, or 20% BSFL. Analysis of Similarities (ANOSIM Bray-Curtis) in broiler caeca community structure.

Figure 5 The microbial composition of 21 (A) and 42 (B) days old broilers ordinated by principal component analysis (PCA) of broilers fed 0%, 5%, 10%, 15%, or 20% BSFL.

The red circle represents the control group, green square represents the 5% BSFL group, the blue rhombi represent the 10% BSFL group, the gray triangle represent the BSFL group, and the yellow upside-down triangle represents the 20% BSFL group in the PCA analysis.

Abundance

A significant reduction was observed in the abundance of sequencing variants representing Enterococcus at the genus level in the caeca of the BSFL-fed broilers at day 21 (p-value = 0.048; Fig. 6A). The highest abundance of Enterococcus was observed in the caeca content of broilers fed 0% BSFL. Furthermore, the abundance of the Unclassified christensenellaceae group was the lowest in the caeca of 21 days old broilers fed 20% BSFL (p-value = 0.025; Fig. 6B).

Figure 6 (A) Abundance of Enterococcus in the caeca of 21 days old broilers fed 0% (red), 5% (green), 10% (blue), 15% (grey), or 20% (yellow) BSFL. (B) Abundance of unclassified Christensenellaceae in the caeca of 21 days old broilers fed 0%, 5%, 10%, 15%, or 20% BSFL.

Abundance plot of bacterial population in 21 days old broilers caeca microbiota of different groups was compared through analysis of variance (ANOVA).

The abundance of diverse bacterial populations at the genus level was not affected by the BSFL dietary inclusion either at day 21 (Fig. 7A) or 42 (Fig. 7B). In the caeca obtained from 42 days old broilers, there was a negative relationship between Coprococcus and BSFL inclusion levels (Fig. 8A; R = 0.326; p-value = 0.043), a positive relationship between the abundance of the genus Roseburia (R = 0.426; p-value = 0.0069), and a positive relationship between Dehalobacterium (R = 0.458; p-value = 0.034) and the BSFL dietary inclusion levels (Figs. 8B and 8C, respectively). The highest abundance of Dehalobacterium was observed in the caeca of broilers fed 20% BSFL, whereas the lowest abundance of this group of bacteria was observed in the caeca of broilers fed 0% BSFL.

Figure 7 Relative abundance of microbiome at genus level in the caeca of 21 (A) and 42 (B) days old broilers.

On top of Figs. 6A and 6B (x-axis) there are red, green, blue, grey, and yellow representing the BSFL percentage groups (0% (red), 5% (green), 10% (blue), 15% (grey), and 20% (yellow)). Different colours in the squares at the legend boxes represent the bacterial groups found in the broilers caeca samples analysed. On the y-axis, the abundance of the bacterial groups found is represented from 0 to 40.

Figure 8 Spearman correlation was performed to evaluate the relative abundance of Coprococcus (A), Roseburia (B), and Dehalobacterium (C).

All three graphs are representing the bacterial abundance increases at genus level. The positive correlation of the Black Soldier Fly larvae (BSFL) with the Roseburia abundance in the caeca microbiome of the 42 days old broiler chickens is most likely linked with a broilers performance improvement.

Discussion

The performance of the broilers (feed intake, body weight gain, and feed conversion ratio) used for this study was improved by the 20% BSFL inclusion through an increase in body weight gain (from 3 kg to 3.25 kg) and a reduction in feed conversion ratio from ~1.5 to ~1.3 (de Souza-Vilela et al., 2021). There is available research demonstrating the impact of dietary insect inclusion on the immune system and positive performance of livestock, pet, and aquatic animals such as chickens, pigs, dogs, and fisheries (de Souza-Vilela et al., 2021; Foysal et al., 2019; Gasco, Finke & Van Huis, 2018; Lei et al., 2019; Roos & Van Huis, 2017; Spranghers et al., 2018). As previously described (de Souza-Vilela et al., 2021), the immune system of broilers could gradually be modulated when broilers were fed up to 20% BSFL, indicated by significantly reduced blood and intestinal intraepithelial lymphocytes at 42 days old. Other investigators have shown that intraepithelial immune cells and performance of broilers can be influenced by altering the GIT microbiome (Rubio et al., 2015; Yang et al., 2019). As there were improvements in the broiler performance and an immune system modulation of the broilers fed up to 20% full-fat BSFL, alterations in the GIT microbiome were expected. However, these effects were only minor both at day 21 and day 42 and can therefore be considered as negligible.

In this study, Shannon diversity values varied from 1.6 to 2.6 in 21-day-old broilers and from 4.4 and 5.6 in 42-day-old broilers. Biasato et al. (2018a) reported a Shannon index of 8 in 97-day-old Label Hubbard hybrid free-range broilers fed insect (Tenebrio molitor) meal. A common variation of Shannon diversity in conventional systems is from 3 to 6 because broilers in conventional systems have lower GIT microbial diversity values compared to free-range broilers (Biasato et al., 2018a). The lower Shannon diversity (from 1.6 to 5.6) found in our study is likely to be due to the housing system used (cages). In addition, investigating these birds at a relatively young age, the broiler’s development of a diverse microbiome is limited. As described in the results section and Fig. 3, younger broilers had a lower cecal microbiome diversity (1.6–2.6; Fig. 3A) compared to older broilers (4.4–5.6; Fig. 3B). There was no difference in alpha diversity (Chao1 and Shannon index) within treatments. There was a slight reduction in the ANOSIM in the caeca microbiome of broilers fed 20% BSFL compared to the control group (400 compared to 500). Benzertiha et al. (2019) and Józefiak et al. (2020) found no differences in microbial alpha diversity in the caeca between 35 days old female floor pen Ross 308 broilers fed insects and broilers from the control group. In contrast, other studies reported a major increase in the caeca diversity caused by dietary insect meal inclusions. For instance, Biasato et al. (2018a) included 7.5% of full-fat Tenebrio molitor in the diets (containing less than 0.04% lauric acid, other antimicrobial compounds are unknown) of 43 days old free-range Label Hubbard broiler chickens fed for 54 days and found higher alpha (Shannon index) and beta diversity (PC2) in the caeca of free-range broiler chickens fed diets containing Tenebrio molitor meal compared to the control group. The discrepancies between these studies and ours are probably not only due to the level of unknown amounts of antimicrobial compounds present in the diets, insect meal, and dietary inclusion level but also because of the poultry breed, age, and environmental conditions as we are comparing results of 35-, and 42-days old Ross 308 broilers in a conventional system with results of 97-days old free-range Label Hubbard broiler chickens. Interestingly, Biasato et al. (2020) found a negative effect in the caeca microbiome diversity of 35 days old broiler chickens of the Ross 708 breed fed a diet containing 15% BSFL (bioactive compounds are unknown) compared to control, 5%, and 10% BSFL fed groups. The 15% BSFL fed group demonstrated a lower Shannon diversity index (6.49) compared to the control (7.25), 5% (6.88), and 10% (7.36) BSFL. These results are probably due to high variability in the gastrointestinal tract of poultry with discrepancies in the poultry breed, insect meal composition, and dietary inclusion levels (Stanley et al., 2013).

Benzertiha et al. (2019) and Józefiak et al. (2020) found no differences in microbial alpha diversity in the caeca between 35 days old female floor pen Ross 308 broilers fed insects and broilers from the control group. In contrast, other studies reported a major increase in the caeca diversity caused by dietary insect meal inclusions. For instance, Biasato et al. (2018a) included 7.5% of full-fat Tenebrio molitor in the diets (containing less than 0.04% lauric acid, other antimicrobial compounds are unknown) of 43 days old free-range Label Hubbard broiler chickens fed for 54 days and found higher alpha (Shannon index) and beta (PC2) diversity in the caeca of free-range broiler chickens fed diets containing Tenebrio molitor meal compared to the control group. The discrepancies between these studies and ours are probably not only due to the level of unknown amounts of antimicrobial compounds present in the diets, insect meal, and dietary inclusion level but also because of the poultry breed, age, and environmental conditions as we are comparing results of 35-, and 42-days old Ross 308 broilers in a conventional system with results of 97-days old free-range Label Hubbard broiler chickens. Interestingly, Biasato et al. (2020) found a negative effect in the caeca microbiome diversity of 35 days old broiler chickens of the Ross 708 breed fed a diet containing 15% BSFL (bioactive compounds are unknown) compared to control, 5%, and 10% BSFL fed groups. The 15% BSFL fed group demonstrated a lower Shannon diversity index (6.49) compared to the control (7.25), 5% (6.88), and 10% (7.36) BSFL. These results are probably due to high variability in the gastrointestinal tract of poultry with discrepancies in the poultry breed, insect meal composition, and dietary inclusion levels (Stanley et al., 2013).

The relationship between bioactive components such as antimicrobial peptides, chitin, and lauric acid and microbial enhancements in insect diets is poorly understood. The lauric acid content in the diets is higher in full-fat insect meals. However, defatted insect meals are never entirely defatted. For instance, the range of variation reported in the literature is 4.6% to 18% fat content in defatted BSFL meals (Renna et al., 2017; Mwaniki et al., 2020; Schiavone et al., 2017). In our study, we used full-fat BSFL with 32% crude fat, including 13% lauric acid. Since it has been previously reported that chitin and the expression of antimicrobial peptides are dependent on the BSFL diet (Spranghers et al., 2017; Vogel et al., 2018), some controversies may be related to the chitin extraction methods used (Soetemans, Uyttebroek & Bastiaens, 2020). We considered that the amount of chitin in the BSFL used in our experiment was around 6.2%, based on the values reported by Spranghers et al. (2017) as the rearing substrate of the BSFL we added in our diets was chicken feed. The chitin content in the diets in this study increased along with the BSFL dietary inclusion from 0 (control) to 1.24% (20% BSFL inclusion). The BSFL antimicrobial peptides (presented in the aqueous extracts) have demonstrated greater power against gram-negative bacteria (E. coli and P. fluoresces) if the BSFL reared on high protein diets and greater power against gram-positive bacteria (M. luteus and B. subtilis) if reared on high chitin or cellulose diets (Vogel et al., 2018). Linking insect antimicrobial peptides’ quantity, type, and their action to the health status and gut microbiome of poultry and other livestock animals is a key future step in this research area.

In this study, the relative abundance of the caeca microbiome at the genus level was compared between control and broilers fed graded BSFL levels. No dietary effect was observed in most bacterial groups at days 21 and 42. However, the inclusion of BSFL in broiler diets decreased the abundance of sequence variants representing Enterococcus (gram-positive group of bacteria) and unclassified Christensenellaceae (gram-negative group of bacteria) in the caeca of 21-days old Ross 308 broilers. On the contrary, Biasato et al. (2020) reported no differences in these variants at the genus level when Ross 708 broilers were fed with 0%, 5%, 10%, and 15% of partially defatted BSFL meal for 35 days. Additionally, Biasato et al. (2018a) did not show any differences either in Enterococcus or unclassified Christensenellaceae bacterial groups between free-range female Hubbard hybrid broiler chickens fed a diet with 7.5% Tenebrio molitor and control. The differences between the two previously mentioned studies and ours are expected as broiler breeds (Ross 708, Ross 308, and Hubbard), environment (conventional system and free-range), insect species, and dietary inclusion differed between studies. The decrease in the variants representing Enterococcus in the caeca of broilers in our study could be a negative or a positive response caused by the BSFL dietary inclusion. Within Enterococcus, there are strains that can be either beneficial probiotics or act highly pathogenic (Bauer et al., 2019). The main Enterococcus species include E. faecium and E. faecalis, which have been recognized as probiotics that improve the immune system and performance of broilers (Royan, 2018).

The bacterial group of the Christensenellaceae family (phylum Firmicutes) and a lower abundance of the genus Dehalobacterium have been associated with a high body mass index in humans (Fu et al., 2015). Li et al. (2018) reported improvement in hens’ intestinal function accompanied by a Dehalobacterium decrease. In the current study, the bacterial group of the Christensenellaceae family reduced and the sequence variants representing the Dehalobacterium increased in the caeca of broilers as the levels of BSFL increased in the diets. Contrary to the previously cited literature, the decrease in unclassified Christensenesenellaceae and Dehalobacterium was accompanied by a broiler body weight increase in our study (de Souza-Vilela et al., 2021). Furthermore, the abundance of Coprococcus presented a negative correlation with the BSFL inclusion levels at day 42. The negative correlation and consequent decrease in the genus Coprococcus upon feeding BSFL are possibly not positive for the gut health of broilers, as this bacterial genus is being reported to be beneficial. The Coprococcus genus has been reported to be involved in butyrate production and, therefore beneficial to broiler performance, intestinal villus structure, and the decrease of pathogenic bacteria (Onrust et al., 2015; Pryde et al., 2002). Biasato et al. (2018a) reported contrasting results in 97 days old Label Hubbard broilers fed 7.5% of the full-fat Tenebrio molitor larvae meal, suggesting a higher population of the Coprococcus in the insect-fed group compared to the control group.

The broiler body weight increase is probably due to the increase in other bacterial groups in the caeca of broilers. For instance, there was a positive correlation of the Roseburia (gram-positive group of bacteria) with the BSFL inclusion. The increase in the abundance of the genus Roseburia is beneficial to the gut health of broilers as this bacterial group has also been reported to be a butyrate producer (Biasato et al., 2020; Scott et al., 2011). Butyrate has been considered to reduce feed conversion ratio, increase body weight, enhance immune response, and improve gut health and gut development (Levy et al., 2015). Our results are in accordance with Biasato et al. (2020), who reported a higher abundance of Roseburia in the caeca of broiler chickens fed 15% BSFL meal compared with the other groups (0%, 5% and 10% BSFL-fed groups).

In conclusion, the modifications in the relative abundance of caeca microorganisms in broiler chickens fed up with 20% BSFL are minor and complex to characterise as positive or negative to the gut health of broilers. The GIT microbiome is variable and there is no specific group of bacteria (gram-positive or gram-negative) that increases or decreases with the dietary insect meal inclusions. Although most of the literature reports positive effects of insect meal dietary inclusions in the performance, health, and GIT microbiome of poultry, further research is necessary to compare the effect of the BSFL inclusion in the microbiome population in different parts of the GIT. The extraction of the BSFL bioactive components such as lauric acid, chitin, and antimicrobial peptides and their effects on the whole GIT of poultry should be performed. Future research studies using the same or similar environmental conditions, poultry age range, and poultry breed are valuable. Further, histology, intraepithelial lymphocytes and GIT microbiome alterations at different dietary inclusion levels of the bioactive compounds extracted from the BSFL would be worth investigating. The quantity and quality of antimicrobial contents of insect meals are known to be diet-dependent, and this field should also be further studied and possibly considered when adding an insect meal into poultry diets.

Conclusions

The present research demonstrates the feasibility of including up to 20% Black Soldier Fly larvae (BSFL) in broiler diets without relevant changes in caeca microbiota. The bacterial diversity was not affected by the BSFL inclusion but decreased the abundance of variants representing Enterococcus and unclassified Christensenellaceae at day 21 of broilers fed BSFL while at day 42 decreased the abundance of Coprococcus and increased the abundance of the Roseburia and Dehalobacterium genus.

Supplemental Information

Supplemental Information 1 Author Checklist.

Click here for additional data file.

Supplemental Information 2 Sequencing Data.

Click here for additional data file.

As well as Professor Robert Swick and the Poultry Research and Teaching Unit at the University of New England with all support and special contribution.

Additional Information and Declarations

Competing Interests

Author Contributions

Animal Ethics

DNA Deposition

Data Availability

The authors declare that they have no competing interests. The authors worked with the following non-academic collaborators: Feedworks (Romsey, VIC, Australia), Karma3 (Thomastown, VIC, Australia), and Go Terra (Canberra, ACT, Australia).

Jessica de Souza Vilela conceived and designed the experiments, performed the experiments, analyzed the data, prepared figures and/or tables, authored or reviewed drafts of the article, and approved the final draft.

Sarbast K. Kheravii performed the experiments, analyzed the data, prepared figures and/or tables, authored or reviewed drafts of the article, and approved the final draft.

Yadav Sharma Bajagai analyzed the data, prepared figures and/or tables, authored or reviewed drafts of the article, and approved the final draft.

Manisha Kolakshyapati performed the experiments, authored or reviewed drafts of the article, and approved the final draft.

Terence Zimazile Sibanda analyzed the data, authored or reviewed drafts of the article, and approved the final draft.

Shu-Biao Wu performed the experiments, authored or reviewed drafts of the article, and approved the final draft.

Nigel R. Andrew conceived and designed the experiments, authored or reviewed drafts of the article, and approved the final draft.

Isabelle Ruhnke conceived and designed the experiments, performed the experiments, authored or reviewed drafts of the article, and approved the final draft.

The following information was supplied relating to ethical approvals (i.e., approving body and any reference numbers):

The University of New England Animal Ethics Committee approved this research (AEC18-084).

The following information was supplied regarding the deposition of DNA sequences:

The sequences are available at NCBI SRA: PRJNA935040.

The following information was supplied regarding data availability:

The data is available at figshare: de Souza Vilela, Jessica (2023). Microbiota raw data.csv. figshare. Dataset. https://doi.org/10.6084/m9.figshare.21975419.v1.

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
