# Peer review of "Inclusion of up to 20% Black Soldier Fly larvae meal in broiler chicken diet has a minor effect on caecal microbiota"

_PeerJ, doi:10.7717/peerj.15857_

## Round 0.1 · original submission · Major Revisions

The reviewers have some serious concerns about the experimental design. Please consider these comments and others regarding the reporting of results in your revision.

·

Basic reporting

1. English can be improved
2. The introduction lacks proper flow. Consider rewriting the Introduction

Experimental design

1.Materials and methods need more details
2. The performance parameters were not reported. There are 5 treatments with 8 replicates and 10 birds per replicate. Yet, only two birds were sampled by the end of the study (one at day 21 and second bird at day 42) for microbiome analysis. The sample size is too small to derive conclusions

Validity of the findings

Results and discussion section is fairly well written

Additional comments

Lines 22-23: It may be true that some of the feed ingredients act as substrate for intestinal microbiota, more specifically some beneficial bacteria utilizing prebiotics in feed, but feeding the microbiota cannot be the sole purpose of feed ingredients. The word intestinal microbiota is too broad that includes both good and bad bacteria. The objective/aim of the study was not stated clearly.
Line 25-26: Is the diet iso-caloric and iso-nitrogenous?
Lines 51-63: Correct the font. This section is in a different font than rest of the manuscript
Lines 51-126: The Introduction section has a bunch of sentences or statements from literature which are not well connected. Some of the statements are more suited for Discussion than Introduction. Be clear and rewrite the Introduction. Provide general information about the study and importance/ justification for the current study.
Lines 58-60: Include appropriate reference.
Line 61: What do the authors mean by waste reduction performance? Clarify
Lines 62-63: Make sure the references were cited as per journal requirements. Check the entire manuscript and correct as needed.
Line 62: correct as ‘on dry matter basis’
Lines 65-73: How do the authors explain the use of a feed ingredient that is not consistent in nutrients and antimicrobial components and its effect on the intestinal microbiome.
Line 89: In vivo and in vitro needs to be italicized. Check the entire manuscript and correct as needed.
Line 89-92: All antimicrobial compounds do not demonstrate prebiotic action. Antimicrobial peptides and lauric acid may have some beneficial effects but not prebiotic action.
Lines 110-122: This sentence is awkward, consider paraphrasing the sentence.
Line 128: Correct as ‘Materials and Methods’
Lines 130-167:
• Was the BSFL analyzed before inclusion in the feed or the values presented were based on available literature?
• Include materials and methods for determining the levels of chitin, lauric acid and antimicrobial peptides in BSFL
• Any preliminary studies done to determine digestibility of BSFL in poultry gut?
• Correct typos and grammatical errors
• Why are the BSFL inclusion levels are different in starter vs grower and finisher diets?
• Were the birds weighed during the study and evaluated for feed intake and body weight gains?
• Is there any difference in the performance parameters between treatments?
Line 207: Is it Analysis of Similarities or dissimilaries? Correct ANOSIM is an acronym, not a word. Correct as ANOSIM
Line247-249: That’s a big claim. Include appropriate references to support such statements.

Reviewer 2 ·

Basic reporting

The current manuscript evaluates the effects of including BSF larvae in broiler chicken diet on their caeca microbiota. Despite the topic of the research being not really innovative, as there are numerous research assessing gut microbiota changes in insect-fed poultry, it still deserves some attention from the scientific community. However, there are serious issues that makes the research not scientifically sound and reliable (please, see the section related to the Experimental design comments). Independently on that, I have some specific comments related to the first sections of the paper.

1) Title: by reading this, it seems that BSF larvae were adiministered to the birds. Please, replace "BSF larvae" with "BSF larva meal".

2) Abstract: it contains all the relevant information.

3) Introduction:
- Some aspects are unnecessary and out of the scope of your researcj (i.e., lines 104-116). The authors did not perform any challenge test and also mentioning the use of B. licheniformis has no relevance, in this context;
- The cited literature is not updated and does not allow to fully contextualize the topic. There are several research in the years 2020-2022 about the use of HI and TM meals in poultry and their impact on gut microbiota that are completely missing here. It is more important, in my opinion, to focus on the impact of insect meals than on chitin and lauric acid taken as single ingredients/additives (i.e., lines 95-102), as the authors worked with a full-fat meal and not with the single nutraceutical components of BSF.

Experimental design

- There is no table reporting the formulation and the nutrient composition of the experimental diets, thus not allowing me to check if they are correct and if the experimental design has sense;
- Why in the starter diet the inclusion levels of BSFL are completely different from grower and finisher diets? It is impossibile to clearly discriminate the effects on gut microbiota changes at 21 and 42 days of age related to the different inclusion levels tested if in the starter period the birds actually received a different amount of insect meal in the different experimental treatments. This represents a fatal flaw;
- The evaluation of the growth performance is missing. The assessment of gut microbiota alone has no sense if not accompanied by the parallel assessment of bird performance, as we are talking about production animals and the gut health is actually a synonymous of animal health and directly impacts animal performance.

Validity of the findings

Please, check my previous comments.

Additional comments

- Results are very synthetic and reported in a too superficial way. A general overview of phyla and genera in all the experimental treatments is missing, as the numbers of reads or any information about the quality of sequencing process.

·

Basic reporting

No comment

Experimental design

No comment

Validity of the findings

No comment

Additional comments

The article is well structured and the English language use is understandable to a large audience, be it a proof reading by a fluent English speaker is recommended to eliminate some minor grammatical errors throughout the text. The discussion and conclusions are well thought through and substantiated by supporting literature.

---

## Round 0.2 · accepted · Accept

Thank you for your revision addressing reviewer comments.